A novel birnavirus identified as the causative agent of summer atrophy of pearl oyster (Pinctada fucata (Gould))

Matsuyama Tomomasa 1 matsuyama_tomomasa55@fra.go.jp
Miwa Satoshi 1
http://orcid.org/0000-0003-0814-6146 Mekata Tohru 1 2
Kiryu Ikunari 1
Kuriyama Isao 3 4
Atsumi Takashi 3
Itano Tomokazu 5
Kawakami Hidemasa 5
1 Japan Fisheries Research and Education Agency, Pathology Division, Aquaculture Research Department, Fisheries Technology Institute , Minami-Ise, Mie , Japan
2 Okayama University of Science, Department of Veterinary Medicine, Faculty of Veterinary Medicine , Imabari, Ehime , Japan
3 Mie Prefecture Fisheries Research Institute , Shima, Mie , Japan
4 Mie Prefectural Government Department of Agriculture, Forestry and Fisheries , Tsu, Mie , Japan
5 Ehime Fisheries Research Center , Ehime , Japan
Breitbart Mya
Electronic publication date: 2024 Apr 30
Publication date: 2024
Volume: 12
Electronic Location ID: e17321
Received 2024 Jan 19; Accepted 2024 Apr 9
Copyright: © 2024 Matsuyama et al.
Copyright year: 2024
Copyright holder: Matsuyama et al.
License: This is an open access article distributed under the terms of the Creative Commons Attribution License, which permits unrestricted use, distribution, reproduction and adaptation in any medium and for any purpose provided that it is properly attributed. For attribution, the original author(s), title, publication source (PeerJ) and either DOI or URL of the article must be cited.
License URL: https://creativecommons.org/licenses/by/4.0/

Keywords: Pinctada birnavirus, Entomoviridae, Birnavirus, Pinctada fucata, Mass mortality, Pearl oyster

Funding: Food Safety and Consumer Affairs Bureau, the Ministry of Agriculture, Forestry, and Fisheries of Japan and JSPS KAKENHI JP 22H02440 This work was supported by the Food Safety and Consumer Affairs Bureau, the Ministry of Agriculture, Forestry, and Fisheries of Japan and JSPS KAKENHI Grant Numbers JP 22H02440. The funders had no role in study design, data collection and analysis, decision to publish, or preparation of the manuscript.

==============================
The Akoya pearl oyster (Pinctada fucata (Gould)) is the most important species for pearl cultivation in Japan. Mass mortality of 0-year-old juvenile oysters and anomalies in adults, known as summer atrophy, have been observed in major pearl farming areas during the season when seawater temperatures exceed about 20 °C since 2019. In this study, we identified a novel birnavirus as the pathogen of summer atrophy and named it Pinctada birnavirus (PiBV). PiBV was first presumed to be the causative agent when it was detected specifically and frequently in the infected oysters in a comparative metatranscriptomics of experimentally infected and healthy pearl oysters. Subsequently, the symptoms of summer atrophy were reproduced by infection tests using purified PiBV. Infection of juvenile oysters with PiBV resulted in an increase in the PiBV genome followed by the atrophy of soft body and subsequent mortality. Immunostaining with a mouse antiserum against a recombinant PiBV protein showed that the virus antigen was localized mainly in the epithelial cells on the outer surface of the mantle. Although the phylogenetic analysis using maximum likelihood method placed PiBV at the root of the genus Entomobirnavirus, the identity of the bi-segmented, genomic RNA to that of known birnaviruses at the full-length amino acid level was low, suggesting that PiBV forms a new genus. The discovery of PiBV will be the basis for research to control this emerging disease.

Introduction

The Akoya pearl oyster (Pinctada fucata (Gould)) is the most important species for pearl culture in Japan and one of the most well-studied animals for biomineralization (Simkiss & Wilbur, 2012). Mass mortality of juvenile pearl oysters has occurred every summer in major pearl farming areas of Japan since 2019 (Matsuyama et al., 2021; Sano, Kuriyama & Komaru, 2021). According to the statistics of the Ministry of Agriculture, Forestry and Fisheries, Japanese pearl production in 2021 was about 60% of that in 2018. Mortality was particularly significant in juvenile oysters, with a survey conducted by Mie Prefecture showing that the annual mortality rate of age-0 juveniles was 56–74% in 2019 compared to about 15% in previous years, and 23–24% for adults in 2019 compared to 9–16% in previous years (Sano, Kuriyama & Komaru, 2021). The soft body of affected individuals is atrophied toward the dorsal margin. Many adults recover from atrophy after developing the disease, but surviving individuals exhibit symptoms of so-called shell disease (Matsuyama et al., 2021; Sano, Kuriyama & Komaru, 2021) with melanin deposition on the shell nacre layer (Sano, Matsuyama & Inoue, 2024). In addition, the normally smooth nacreous layer is often raised along the edge of the atrophied mantle in surviving oysters. The expression pattern of genes involved in shell formation is altered in the mantle of individuals with atrophied soft body (Sano, Kuriyama & Komaru, 2021; Sano, Matsuyama & Inoue, 2024).

In our previous report (Matsuyama et al., 2021), we characterized the disease and reported it as “summer atrophy”. We further elucidated that this condition is an infectious disease caused by an unknown pathogen; the pathogen was not inactivated by either the filtration through a 100 nm filter or by treatment with chloroform, suggesting that the causative agent is a virus less than 100 nm in diameter with no envelope (Matsuyama et al., 2021). However, the etiological virus responsible for summer atrophy has not been clarified.

In this study, we conducted a series of experiments (Fig. 1) and identified a novel birnavirus as the causative agent of summer atrophy and named it Pinctada birnavirus (PiBV).

Figure 1 Outline of the process of the present study to identify the causative agent for summer atrophy of the pearl oyster.

First, infection source, which was supposed to contain the etiological agent, was prepared by a preliminary experimental infection using spontaneously affected oysters as the donors along with the negative control inoculum (#1). The presence or the absence of the pathogen in the inocula was confirmed by an experimental infection. Using the prepared inocula, an experimental infection of the disease was conducted (#2). Then, the infected oysters and control oysters were compared by metatranscriptomics (#3) to infer the causative virus by identifying viral genes that appear only in the infected animals. RT-qPCR was constructed (#4) on the basis of the obtained genomic information of the presumed causative virus (PiBV). Subsequently, the virus was purified (#6) from the experimentally infected oysters (#5). Viruses contained in the purified fraction and the negative control inoculum were compared by next generation sequencing (#7), while the pathogenicity of the purified fraction was tested by an experimental infection (#8). Phylogenetic analysis and genomic characterization of PiBV were carried out with the obtained genomic information (#9). The pearl oysters collected from various oyster farms with different histories of summer atrophy were checked for PiBV by RT-qPCR (#10). The changes in the amount of PiBV genome were investigated (#11) in the specimens sampled during the experimental infection (#2). Finally, the localization of PiBV in the pearl oyster tissues was studied by immunohistochemistry (#12) and the pathogenesis of summer atrophy was discussed. *, †: The infection source and negative control inoculum were also used as the positive and negative controls in the infection test (#7) to presume the pathogenesis of summer atrophy.

Materials and methods　

Animals

The pearl oysters used in this study were as follows.

(1) A total of 1-year-old oysters affected by summer atrophy from a pearl farm in Ago Bay, Mie prefecture. The shell width was 58–66 mm, with the average and standard error of 61.4 ± 1.4 mm when randomly measured on five specimens. The soft body was clearly atrophied after a spontaneous disease outbreak. Hemolymph was collected upon arrival at the laboratory of Fisheries Technology Institute in Minami-Ise.

(2) Healthy 4-month-old oysters inseminated and reared in a tank at Mie Prefecture Fisheries Research Institute. The shell width was 3.3–6.1 mm, with the average and standard error of 4.8 ± 0.2 mm when randomly measured on 20 specimens. These animals were used as the recipients of infection studies for comparative metatranscriptomics.

(3) Healthy 3-year-old oysters reared in summer atrophy-free area. Shell width was 74–82 mm, with the average and standard error of 78.4 ± 4.2 mm when randomly measured on five specimens. These were used for infection tests, preparation of infection sources, and virus purification.

The pearl oysters were reared in 65 L tanks filled with 56 L of running sea water (approximately 250 mL/min) filtered through a 1 μm-pore size filter and maintained at 23–25 °C under natural photoperiod in the laboratory of Fisheries Technology Institute. For food, 500 mL of cultured diatoms (Chaetoceros neogracile, approximately 5 × 105 cells/mL) were added to each tank five times a week, both during the acclimation period and during infection tests.

Infection source and negative control inoculum

The infection source and negative control inoculum were prepared using experimentally infected or seawater-injected pearl oysters. Healthy 3-year-old oysters were divided into two groups of three individuals each. One group was inoculated with pooled hemolymph collected from the adductor muscles of six 1-year-old oysters affected by summer atrophy, and the other with sterile seawater (500 μL/individual) and kept in running water at 25 °C for 5 days. The extrapallial fluid and soft body of the three individuals of each group were collected in separate beakers, and the soft body was cut into small pieces with scissors. These were filtered through a 100 μm-pore nylon mesh to remove tissue fragments, and the filtrates were centrifuged at 500 × g for 10 min. The supernatant of each filtrate was further centrifuged at 8,100 × g for 10 min to remove particulate matter. To this supernatant, penicillin G and streptomycin sulfate were added at final concentrations of 1,000 units/mL and 1,000 μg/mL, respectively. This was filtered through a 0.22-μm filter (Thermo Fisher Scientific, Waltham, MS, USA), and stored at 4 °C as the infection source or negative control inoculum until use. These were used for the experimental infections for comparative metatranscriptomics and virus purification. A portion of each inoculum was stored at −80 °C and later quantified for the copy numbers of a PiBV gene. To confirm the presence of the pathogen in the infection source and the absence in the negative control inoculum, each inoculum was injected into the adductor muscle of five, healthy 3-year-old oysters (200 μL/oyster), and the mortality of the oysters were monitored for 3 weeks. The extent of abnormalities formed in the nacreous layer of surviving animals was quantitatively evaluated according to the shell score used in a previous paper (Matsuyama et al., 2021). That is, score 1 was defined as an abnormal area of less than 30%, score 2 as an abnormal area greater than 30% but less than 70%, and score 3 as an abnormal area of greater than 70%. The shell score for an individual oyster was calculated by adding the scores of the two valves of the animal.

Experimental infection for comparative metatranscriptomics

One healthy, 4-month-old oyster was placed in each well of eight 24-well plastic plates (Thermo Fisher Scientific, Waltham, MA, USA), and each well was filled with 1 mL of autoclaved seawater. Thus, a total of 192 oysters were used. The plates were divided into two groups, with four plates designated as the infection group and the remaining four as the control group. Either the infection source or negative control inoculum was added to each group (200 µL/well) and the plates were incubated at 25 °C for 2 h to establish infection. Oysters in the wells were then washed three times with autoclaved seawater, and each well was filled with 1 mL of seawater. The oysters in the plates were reared at 25 °C in the dark without feeding. For both infection and control groups, two plates were designated as observation plates, and the other two as sampling plates. The observation plates were examined daily using a stereomicroscope (Leica MZ16, Wetzlar, Germany) to observe atrophy through translucent shells and to count dead oysters. Apart from the oysters incubated in the well plates, four individual oysters were sampled as the initial controls before infection. After the start of the experiment, four oysters were randomly taken from sampling plates of both groups on days 2, 4, 8, 14, and 21. The sampled animals were frozen in liquid nitrogen and stored at −80 °C for later analyses. On days 2 and 4 after infection, four individuals from each sampling plate were taken from both groups, fixed in Davidson’s solution (Bell & Lightner, 1988). The fixed samples were dehydrated through an ethanol series and embedded in paraffin for histological analyses.

Comparative metatranscriptomics

The comparative metatranscriptomics was conducted on the specimens that were cryopreserved as stated above. Total RNA was extracted from each of 4 oysters sampled from each group on either day 2 or day 4 using TRIzol LS (Thermo Fisher Scientific, Waltham, MA, USA) and dissolved in nuclease-free water. The RNA samples thus obtained from four individuals sampled at the same time in each test group were pooled to create four RNA samples. The RNA was further purified using Nucleospin RNA XS (Takara Bio, Shiga, Japan). Library preparation and sequencing were conducted by Bioengineering Lab. Co., Ltd (Kanagawa, Japan). After removing ribosomal RNA using MGI Easy rRNA Depletion Kit (MGI Tech, Shenzhen, China), libraries were prepared according to the manufacturer’s instructions using MGI Easy RNA Directional Library Prep Set (MGI Tech, Shenzhen, China). Reads were obtained using the DNBSEQ-G400 system (MGI Tech, Shenzhen, China) at 2 × 200 bp. CLC Genomics Workbench ver. 21.0.5 (Qiagen, Hilden, Germany) was used for subsequent data analyses. Host-derived reads were removed by mapping the reads against the genome sequence of the pearl oyster (Takeuchi et al., 2016; Takeuchi et al., 2022). The reads that did not match the oyster genome were assembled to contigs. The contig sequences longer than 500 bp were subjected to homology analyses (Blastx) against registered viral sequences. For blast analysis, all viral identical proteins (1,797,225) in GenBank (as of October 14, 2021) were downloaded and used them as a database for CLC’s local blast. The numbers of reads mapped to the contigs showing homology to known viral sequences were counted for each of the four RNA samples. For the contigs that did not show homology to known viral sequences, we attempted to select contigs that met the following three conditions as candidates for pathogen-derived sequences. (1) The sequence is more than 500 bp. (2) More than 200 reads were mapped on the contig. (3) The reads only from the infection group were mapped on the contig. However, of the contigs greater than 500 bp that showed no similarity to known viral sequences, there were no contigs that had more than 200 mapped reads derived only from the infection group.

Construction of reverse transcription quantitative PCR

Since substantial amounts of the genes belong to a certain birnavirus genome were found only in the infected oysters with the comparative metatranscriptomics, we designated the virus as PiBV, as the putative causative agent of summer atrophy. To quantify the PiBV genome, we designed specific primers and a probe (Table 1) for RT-qPCR using primer3 (v. 0.4.0) (Untergasser et al., 2012; Koressaar & Remm, 2007) based on the segment A sequence of PiBV. The total RNA was extracted from samples using TRIzol LS and dissolved in nuclease-free water. The PiBV genome was quantified using THUNDERBIRD Probe One-step qRT-PCR Kit (Toyobo, Osaka, Japan). The reaction mixture was prepared according to the manufacturer’s instructions with 2 μL of total RNA as a template, and LightCycler 96 (Roche, Basel, Switzerland) was used for the measurement. The RT-qPCR cycle was 50 °C for 10 min, 95 °C for 60 s, followed by 40 cycles of 95 °C for 15 s and 60 °C for 45 s. Standard curves were constructed using pCR2.1 TOPO vector (Thermo Fisher Scientific, Waltham, MA, USA) containing the segment A fragment (Table 1). The number of the PiBV gene in an individual specimen was estimated from the number of the virus gene detected, the amount of RNA extracted from each oyster, and the amount of RNA loaded to the reaction mixture.

Table 1 Primers and probes used in this study.

Primer or probe	Sequence (5′-3′)	Usage	
PiBV-qF	ACAATCCAAGGTGCTATCG	RT-qPCR primer	
PiBV-qR	ACACAGTCATGTCGCCCATA	RT-qPCR primer	
PiBV-qHEX*	AAGTCCCAGCTGCCAGACTA	RT-qPCR probe	
VP2-inf-F†	gagggatccgaattcATGACAGACACTCAAAACAAACACAACATG	Cloning of VP2 fragment	
VP2-inf-R†	ttaagcagagattacGAGCTTCTGGTAGTCTCCCCCTGTCATAAGG	Cloning of VP2 fragment	
Notes:

* Labelled with a HEX at 5′-terminus and with a BHQ at 3′-terminus.

† Lower case letters are tag sequences for in-fusion cloning.

Virus purification

To prepare the source for purification of PiBV, 200 μL of the infection source prepared as described in “Infection source and negative control inoculum”, was injected into the adductor muscles of 27 healthy 3-year-old pearl oysters (Fig. 1, #5). The animals were reared in flowing water at 25 °C for 3 days. Then, the soft body and body fluid were collected from these oysters, and the soft body was shredded with scissors. The soft body and body fluid were pooled and filtered through a 100 μm nylon mesh. Subsequently, the filtrate was mixed with one-fifth volume of Vertrel XF (Mitsui Chemicals, Tokyo, Japan), and centrifuged at 8,100 × g for 10 min. The supernatant was collected and the Vertrel XF treatment and centrifugation process were repeated two more times. The resultant supernatant was filtered through a 0.22 μm filter, layered onto 60–20% CsCl in TN buffer (10 mM Tris-HCl, 0.85% NaCl pH 7.8), and centrifuged at 110,000 × g for 3 h using an SW32Ti rotor (Beckman Coulter, Brea, CA, USA) at 4 °C. The interface was collected and adjusted to 50 mL with 40% CsCl and then centrifuged at 300,000 × g for 16 h using an NVT65 rotor (Beckman Coulter, Brea, CA, USA) at 4 °C. The samples were collected in 500 µL each from the top layer to the bottom of the tube to make 22 fractions. The PiBV genome was measured by RT-qPCR, and the fraction with the highest concentration of the virus genome was dialyzed against autoclaved seawater and concentrated with PEG 2000. The concentrated sample was layered onto a 20–80% sucrose continuous gradient in TN buffer and centrifuged at 100,000 × g for 14 h using an SW40 rotor (Beckman Coulter, Brea, CA, USA) at 4 °C. The samples were collected in 1 mL each from the top layer to the bottom of the tube to make 12 fractions. The fraction containing the highest amount of the PiBV gene was dialyzed against autoclaved seawater and used as the purified fraction. Negatively stained samples of the purified fraction were prepared for electron microscopy as follows. The purified fraction, either directly or diluted with TN buffer at 1/10, was applied onto the copper grid coated with Collodion membrane (Nisshin EM Co., Ltd. Tokyo, Japan). After the excess sample solution was blotted, the grid was stained with 2% uranyl acetate in 50% ethanol for 5 s. The staining solution was promptly blotted off and the grid was dried and observed with the electron microscope (JEM 1010, JEOL, Tokyo, Japan).

Virus composition of purified fraction and negative control inoculum

The purity of the virus in the purified fraction and the presence of the virus in the negative control inoculum were analyzed (Fig. 1, #7). RNA and DNA were isolated using NucleoSpin RNA/DNA Buffer Set (Takara Bio, Shiga, Japan) from the purified fraction and the negative control inoculum. Libraries were prepared using an MGI Easy RNA Directional Library Prep Set and sequenced on a DNBSEQ-G400 under the conditions of 2 × 200 bp. The sequences from the purified fraction were assembled to construct contigs after removing host-derived reads as described in “Comparative Metatranscriptomics”. The number of mapped reads was counted for each contig and each contig was subjected to homology analysis (BlastX) for viruses. For the negative control inoculum, the obtained reads were mapped to the contig sequences obtained from the purified fraction, and the number of reads mapped to each contig was counted. Data analysis was performed using CLC Genomics Workbench ver. 21.0.5.

Infection test of purified fraction

The pathogenicity of the purified fraction was tested by injection challenge. The purified fraction was diluted 100-fold with autoclaved seawater and used as the inoculum. The infection source and negative control inoculum described in “Infection source and negative control inoculum” were used as the positive and negative control inoculum, respectively. Each of these inocula was injected (100 µL/individual animal) into the adductor muscle of healthy 3-year-old oysters. Test groups were duplicated for the challenge with either purified fraction or the infection source, while a single group was set for the negative control. Ten pearl oysters were used in each group. Oysters were fed and reared in flowing water at 23–25 °C for 3 weeks to observe the mortality. The degree of melanin deposition in the nacre surface of survivors was quantitatively evaluated by the method described above.

Phylogenetic analysis and genomic characterization of PiBV

Alignments of complete amino acid sequences of segment A (polyprotein precursor pVP2-VP4-VP3) and segment B (RNA-dependent RNA polymerase (RdRp)) from birnaviruses were created using ClustalW in MEGA version 11 (Tamura, Stecher & Kumar, 2021). Phylogenetically informative sites were selected using Gblocks 0.91 (Talavera & Castresana, 2007), with the following parameters: “Minimum Number of Sequences for a Conserved Position” was 12, “Minimum Number of Sequences for a Flank Position” was 20, “Maximum Number of Contiguous Nonconserved Positions” was eight, “Minimum Length of a Block” was 10, “Allowed Gap Positions” was “None”, and “Use Similarity Matrices” was “Yes”. The resulting alignments comprising 296 amino acids for segment A and 245 amino acids for segment B were used to infer phylogenetic relationships with MEGA 11 using the Maximum Likelihood method with the LG+G+I model, selected as the best substitution model based on Akaike information criterion (AIC). The phylogenetic robustness of each node was determined by 1,000 bootstrap replicates. The identity and similarity of the full-length amino acid sequences for both segments were calculated using the Smith-Waterman algorithm with EMBOSS water (Madeira et al., 2022). Conserved protein domains present in the PiBV genome were identified using the NCBI conserved domain search (Wang et al., 2023).

Epidemiological investigation

Pearl oysters used for the epidemiological investigation were listed in Table 2. Oysters that experienced summer atrophy were provided by pearl farmers. Oysters were also collected from areas where the summer atrophy had not been reported. RNA was extracted using Trizol LS, and the prevalence of PiBV was analyzed by RT-qPCR.

Table 2 Pearl oysters used for epidemiological investigation for PiBV and results of RT-qPCR.

					Results of RT-qPCR	
Sampling date	Location	Age	Situations of the pearl oysters at the time of sampling	Tissues subjected to RT-qPCR for PiBV	Nomber of oysters tested	Nomber of positive oysters	Rate of positive oysters (%)	
2019.07	Nagasaki	5 month	Survivors after mass mortality	Whole body	8	8	100.0	
2019.08	Mie	5 month	Soft body atrophied	Whole body	18	7	38.9	
2019.08	Kumamoto	1 year	Soft body atrophied	Gill and midgut	9	4	44.4	
2019.08	Ehime	5 month	Survivors after mass mortality	Whole body	9	5	55.6	
2019.08	Nagasaki	2 year	Survivors after mass mortality	Gill and midgut	7	1	14.3	
2019.09	Kumamoto	6 month	During disease outbreak	Whole body	13	12	92.3	
2020.09	Mie	6 month	Survivors after mass mortality	Whole body	25	22	88.0	
2021.07	Nagasaki	5 month	Survivors after mass mortality	Whole body	8	5	62.5	
2021.08	Mie	5 month	During disease outbreak	Whole body	10	10	100.0	
2022.06	Mie	1 year	Soft body atrophied	Hemolymph	11	11	100.0	
2019.08	Ehime	1 year	No abnormality	Gill and midgut	8	0	0.0	
2019.08	Ishikawa	>1 year*	No abnormality	Gill and midgut	10	0	0.0	
2019.09	Wakayama	>1 year*	No abnormality	Gill and midgut	10	0	0.0	
2020.09	Kanagawa	1 year	No abnormality	Mantle, gill and midgut	10	0	0.0	
2022.08	Mie	1 year	No abnormality	Mantle, gill and midgut	10	0	0.0	
Notes:

* Age unknown as they were wild oysters.

The specimens described above the dotted line were sampled from the areas where outbreaks of summer atrophy had been confirmed. The specimens below the dotted line were sampled from the areas where the disease had not been reported.

Dynamics of PiBV in experimental infection for comparative metatranscriptomics

The PiBV genome was quantified by RT-qPCR for the oysters stored at −80 °C in the experimental infection for comparative metatranscriptomics (Fig. 1, #2).

Immunohistochemistry

A mouse antiserum was prepared against a recombinant PiBV protein expressed in Escherichia coli, as follows. The region from the putative start codon to residue 419 of the PiBV segment A was amplified by PCR using specific primers (Table 1) and KOD-plus-neo polymerase (Toyobo, Osaka, Japan). The process from cloning of the amplified product to the preparation of the mouse antiserum was principally the same as that for the preparation of the mouse antiserum against the major capsid protein (MCP) of abalone asfa-like virus (AbALV) (Matsuyama et al., 2021), which was used as the negative control for immunostaining in this study. Briefly, the PCR-amplified product was cloned into the pCold2 expression vector (Takara Bio, Shiga, Japan), and the recombinant protein with an N-terminal His-tag was expressed in E. coli JM109. A mouse was initially immunized with the injection of purified recombinant protein and Freund’s complete adjuvant, and further injected with the recombinant protein and incomplete adjutant three times at 2-week intervals before the serum was collected. For immunohistochemical staining, 3 μm-thick sections were cut perpendicular to the shell surface at several different places from the paraffin blocks prepared from akoya oysters sampled in the experimental infection for comparative metatranscriptomics (Fig. 1, #2) as described above. Sections were triplicated for the day 2 samples and duplicated for day 4 samples. Two specimens, one from the infected and the other from the control group, sampled on day 2 were used up to find optimal experimental conditions, in which the range of the primary antibody was too high (1/100–1/1,000 dilution), resulting in strong non-specific background signals. These were not included in the results of immunohistochemistry (IHC). One specimen of the infected group collected on day 4 was found to be dead (empty shell) and one specimen of the control group of day 2 was accidentally lost. One of the triplicated sections of a day 2 specimen was used as the negative control incubated with the anti-AbALV serum instead of the primary antiserum against PiBV protein. Therefore, the numbers of samples observed with IHC were three for day 2- and day 4-samples of the infected group, two for day 2-samples of the control group, and four for day 4-samples of the control group. One set of sections from each sample was subjected to IHC, and another was stained with hematoxylin and eosin. IHC was performed with some modifications to the method described previously (Matsuyama et al., 2021). The sections were incubated with the mouse anti-PiBV antiserum diluted 1/5,000 in 1% BSA/phosphate-buffered saline at room temperature for 2 h. Sections were then incubated with the peroxidase-labeled secondary antibody (Histofine Simple Stain MAX-PO, Nichirei Biosciences, Tokyo, Japan) for 1 h. Finally, sections were incubated with Histofine Simple Stain AEC solution (Nichirei Biosciences, Tokyo, Japan) for 3 min for color reaction and counterstained with hematoxylin.

Statistical analysis

Results were expressed as means ± standard error. Mortality and prevalence of nacre abnormalities were analyzed statistically with Fisher’s exact test, and the mean shell scores were tested by Mann–Whitney U-test or Steel-Dwass test. A value of p < 0.05 was considered statistically significant.

Results

Pathogenicity of infection source and negative control inoculum

Three oysters injected with the infection source died within 3 weeks, and the surviving two oysters exhibited melanin deposition in their nacre surface, with scores of 1.5 and 3.0, respectively. No deaths occurred in the group injected with the negative control inoculum, and the nacre of the five surviving oysters appeared normal. The results confirmed the presence of the pathogen in the infection source. The copy number of PiBV gene in the infection source was 3.58 × 105/mL by RT-qPCR, whereas that in the negative control inoculum was less than the limit of detection.

Experimental infection for comparative metatranscriptomics

Atrophy was observed only in the infection groups from 3 to 14 days after inoculation. In atrophic animals, the soft body including the mantle contracted toward the hinge, which was consistent with the characteristics of summer atrophy (Fig. 2). Death occurred between 11 and 18 days after inoculation (Fig. 3). In contrast, no atrophy was observed in the control groups, although two animals died in one of the two control groups.

Figure 2 Juvenile pearl oysters at six days after infection.

White and blue dotted lines represent the outer edges of the mantle and gills, respectively. The left panel shows the atrophied soft body, which has shrunken toward the dorsal margin, of a pearl oyster in one of the infection groups, whereas the mantle extends to the edge of the shell to cover the gill in a control animal shown in the right panel.

Figure 3 Changes in the numbers of atrophied oysters (A) and cumulative mortality (B) in the two infection groups and two control groups in the experimental infection for comparative metatranscriptomics.

Each group comprised 24 oysters (of 4-month-old with the average shell width of 4.8 mm). Each oyster was exposed to either the infection source or negative control inoculum for 2 h. (A) Black bars, infection group 1; gray bars, infection group 2. No atrophied animals were observed in the control groups. (B) Black circle, infection group 1; gray circle, infection group 2; black triangle, control group 1; gray triangle, control group 2. *: Significantly different compared to both control groups. Fisher’s exact test p < 0.05.

Comparative metatranscriptomics

In the experimental infection, atrophy started to occur on day 3 (Fig. 3), suggesting that the onset of the pathogen proliferation took place on day 3 or earlier. Therefore, comparative metatranscriptomics was performed on the frozen oysters sampled on days 2 and 4 from the infection and control groups. Approximately 128 million reads were obtained from the four samples. After removing the reads mapped to the genome sequence of the pearl oyster, 161,580 contigs were assembled. The contigs greater than 500 bp were analyzed by BlastX against registered viral sequences. Among them, 27 contigs showed similarity to known viral sequences (Table 3).

Table 3 List of contigs with the best blast hits against known viral sequences.

Accessin no.	Contig size (bp)	Best BlastX hits against virus database	Transcripts per million (TPM)	
				Infection group	Control group	
Blast hit	Viral family	GenBank acc. no.	Identity (%)*	Day 2	Day 4	Day 2	Day 4	
ICTG01000001	3,752	Eridge virus (segment A)	Birnaviridae	AMO03242	38.1	108,643	22,344	0	0	
ICTG01000002	3,172	Eridge virus (segment B)	Birnaviridae	AMO03243	40.7	93,412	18,251	0	0	
ICTG01000003	10,207	Bivalve RNA virus G3	Picornavirales	YP_009329821	26.7	654,888	749,091	797,910	680,038	
ICTG01000004	5,315	Beihai tombus-like virus 18	Unclassified RNA virus	YP_009336951	32.4	90,571	126,718	120,098	92,255	
ICTG01000005	9,577	Barns Ness breadcrumb sponge picorna-like virus 3	Picornavirales	ASM94050	33.1	8,116	13,678	11,161	7,117	
ICTG01000006	3,868	Chaetoceros protobacilladnavirus 4	Bacilladnaviridae	YP_004046697	36.0	11,476	14,588	15,875	9,153	
ICTG01000007	1,300	ssRNA phage SRR6960509_5	Fiersviridae	DAD50998	58.8	4,252	930	15,162	7,126	
ICTG01000008	1,752	Beihai tombus-like virus 17	Unclassified RNA virus	YP_009336793	45.3	4,634	4,601	5,842	11,142	
ICTG01000009	4,253	Beihai tombus-like virus 17	Unclassified RNA virus	YP_009336792	26.9	5,401	9,287	6,863	14,392	
ICTG01000010	1,629	Wenzhou picorna-like virus 14	Unclassified RNA virus	YP_009337718	26.8	2,015	3,464	1,396	1,625	
ICTG01000011	2,423	Picornavirales N_OV_080	Picornavirales	ASG92525	29.2	2,566	3,660	3,911	3,141	
ICTG01000012	3,986	Chaetoceros tenuissimus RNA virus 01	Marnavirida	YP_009505621	28.2	3,467	10,011	4,089	3,984	
ICTG01000013	1,110	ssRNA phage SRR6960509_5	Fiersviridae	DAD50998	53.5	1,400	1,089	9,562	3,577	
ICTG01000014	606	Wenzhou picorna-like virus 17	Unclassified RNA virus	YP_009337186	44.9	570	0	0	0	
ICTG01000015	813	Beihai picorna-like virus 49	unclassified RNA virus	YP_009333504	42.6	1,912	2,975	2,797	814	
ICTG01000016	1,631	Picornavirales sp.	Picornavirales	QRG24245	31.8	1,059	1,730	2,556	406	
ICTG01000017	857	Wenzhou picorna-like virus 16	Unclassified RNA virus	YP_009336962	39.2	1,008	3,762	0	41,695	
ICTG01000018	544	Picornavirales N_OV_080	Picornavirales	ASG92525	37.0	635	0	0	0	
ICTG01000019	621	Wenzhou picorna-like virus 16	Unclassified RNA virus	YP_009336962	38.2	556	1,947	0	43,689	
ICTG01000020	608	Hubei toti-like virus 17	Unclassified RNA virus	YP_009336917	38.5	852	0	0	544	
ICTG01000021	1,131	Picornavirales N_OV_080	Picornavirales	ASG92525	28.9	916	2,495	2,346	78,400	
ICTG01000022	515	Sanxia noda-like virus 1	Unclassified RNA virus	APG76452	38.8	671	0	0	0	
ICTG01000023	629	Wenzhou picorna-like virus 17	Unclassified RNA virus	YP_009337186	34.1	275	1,282	0	526	
ICTG01000024	879	Wenzhou picorna-like virus 32	Unclassified RNA virus	YP_009337247	34.5	393	1,376	431	376	
ICTG01000025	554	Wenzhou picorna-like virus 27	Unclassified RNA virus	YP_009336706	69.9	312	1,455	0	0	
ICTG01000026	871	Gaeavirus sp.	Mimiviridae	AYV80167	51.7	0	2,314	0	0	
ICTG01000027	546	Heterosigma akashiwo virus 01	Phycodnaviridae	YP_009507580	65.0	0	2,953	0	0	
Note:

* Identity of amino acid sequences.

Two contigs (Acc. No. ICTG01000001 and ICTG01000002) that showed homology to the bi-segmented genomes of Eridge virus (Birnaviridae), were formed solely from the reads derived from the infection group and had relatively high numbers of mapped reads (Table 3). Birnaviruses are bi-segmented double-stranded RNA viruses (Delmas et al., 2019). The two contigs were presumed to correspond to the two segments of a single Birnavirus genome, as the transcripts per million (TPM) to these two contigs were roughly equal on both 2- and 4-days post-infection (Table 3). Therefore, a birnavirus (hereafter referred to Pinctada birnavirus or PiBV) possessing the genome of these two contig sequences was considered to be most likely the causative agent of summer atrophy.

Virus composition of purified fraction and negative control inoculum

DNA analysis was not performed because not enough DNA was recovered from the purified fraction. Of the reads obtained by NGS, the sequences matching the pearl oyster genome were excluded. For the rest of the reads, 59.7% and 37.3% formed contig ID-1 and ID-2 shown in Table 4, respectively. The two contigs corresponded to PiBV segments A and B, respectively. In other words, a total of 97.0% of the reads were mapped to the PiBV genome. A small number of sequences obtained from the purified fraction showed similarity to other viruses, such as picornaviruses and unclassified RNA viruses. No NGS reads obtained from the negative control inoculum were mapped to the contigs correspond to the two segments of PiBV (ID-1 and -2). On the other hand, reads from the negative control inoculum were mapped to 23 of the 34 non-PiBV virus contig sequences (ID-3~36) detected from the purified fraction.

Table 4 Data analyses of next-generation sequencing for the purified virus and negative control inoculum.

Contig ID	Occurrence in the purified virus (%)*	Number of mapped reads in negative control inoculum†	Best BlastX hits against virus database	
Blast hit	Blast hit viral family	GenBank acc. no.	Identity
(%)‡	
1§	59.673	0	Eridge virus (segment A)	Birnavirus	AMO03242	38.1	
2||	37.309	0	Eridge virus (segment B)	Birnavirus	AMO03243	37.1	
3	1.868	81	Chaetoceros species RNA virus 02	Picornavirales	YP_010084314.1	81.2	
4	0.984	1,665	Beihai weivirus-like virus 16	Unclassified RNA viruses	YP_009336992.1	26.5	
5	0.020	27,505	Wenzhou picorna-like virus 16	Picornavirales	YP_009336962.1	33.1	
6	0.019	0	Rhizoctonia solani dsRNA virus 10	Unclassified dsRNA viruses	QDW81304.1	28.7	
7	0.017	17,537	Wenzhou picorna-like virus 16	Unclassified Riboviria	YP_009336962.1	33.3	
8	0.013	1,800	No hit				
9	0.012	8,640	Siphoviridae sp.	Siphoviridae sp.	DAL40648.1	50.2	
10	0.010	0	No hit				
11	0.010	5	Marnaviridae sp.	Picornavirales	QJI53791.1	28.4	
12	0.008	326	Fadolivirus 1	Unclassified viruses	QKF94107.1	53.1	
13	0.005	2,213	Siphoviridae sp.	Siphoviridae sp.	DAV83555.1	35.3	
14	0.005	0	Piscine myocarditis virus	Unclassified Totiviridae	AGA37463.1	26.7	
15	0.003	0	Trichoplusia ni TED virus	Riboviria	YP_009507248.1	38.6	
16	0.003	5	Emiliania huxleyi virus 202	Nucleocytoviricota	AET42460.1	49.4	
17	0.003	47	Uncultured Caudovirales phage	Caudovirales	CAB4141778.1	36.1	
18	0.003	0	Partitiviridae sp.	Partitiviridae sp.	QDH91349.1	79.5	
19	0.003	0	Pithovirus LCPAC201	Pithoviridae	QBK90826.1	31.2	
20	0.002	2,669	Enterobacteria phage P7	Caudovirales	YP_009914485.1	99.6	
21	0.002	32	No hit				
22	0.002	2,475	Turtle fraservirus 1	Turtle fraservirus 1	UKB93130.1	24.5	
23	0.002	0	Branchiostoma lancelet adintovirus	Adintoviridae	DAC80271.1	54.9	
24	0.002	66	Gordonia phage Sixama	Siphoviridae	QGF20313.1	38.2	
25	0.002	172	No hit				
26	0.002	239	No hit				
27	0.002	0	Siphoviridae sp. ctBLh2	Siphoviridae sp.	DAF45229.1	26.6	
28	0.002	1,852	No hit				
29	0.001	203	Branchiostoma lancelet adintovirus	Adintoviridae	DAC80271.1	50.5	
30	0.001	56	Terrestrivirus sp.	Megaviricetes	AYV76070.1	51.6	
31	0.001	0	No hit				
32	0.001	0	Carp edema virus	Nucleocytoviricota	BCT22562.1	36.3	
33	0.001	0	Siphoviridae sp.	Siphoviridae	DAN98122.1	20.4	
34	0.001	3,836	Myoviridae sp.	Caudovirales	DAH68598.1	28.9	
35	0.001	8	No hit				
36	0.001	17	Galinsoga mosaic virus	Tombusviridae	NP_044732.1	41.1	
Notes:

* Occurrence = (number of reads mapped to the contig) × 100/(number of total reads used to form contigs).

† Number of reads obtained from negative control inoculum that were mapped to each contig formed from purified virus.

‡ Amino acid identity.

§ Amino acid identity with PiBV segment A (ICTG01000001) is 100%.

|| Identity of amino acid sequence with PiBV segment B (ICTG01000002) is 100%.

Infection test of purified fraction

Bands were not visible in the fraction where the PiBV was supposed to be concentrated in the ultracentrifugation tubes. In the infection test, mortality was low for the oysters injected with the purified fraction, as well as for the oysters of positive control groups, although most of the survivors of these groups exhibited melanin deposits on the nacre (Table 5). The amount of PiBV gene measured by RT-qPCR in the purified fraction used as the inoculum was 4.6 × 104 copies/individual, while that for the positive controls was 7.8 × 104 copies/individual. The PiBV gene was not detected in the inoculum for the negative controls.

Table 5 Results of experimental infection using the purified fraction.

	Purified virus-1	Purified virus-2	Positive control-1	Positive control-2	Negative control	
Mortality (dead/examined)	0/10	1/10	1/10	0/10	0/10	
Nacre abnormalities*	9/10	8/9	8/9	8/10	0/10†	
Mean shell scores of survivors	2.9 ± 0.7	3.6 ± 0.6	2.1 ± 0.6	2.9 ± 0.7	0 ± 0‡	
Notes:

* The number of oysters in which melanin deposition on the nacre surface were observed/the number of surviving oysters.

† Significantly different from the other groups (Fisher’s exact test p < 0.05).

‡ Significantly different from the other groups (Mann–Whitney U-test p < 0.05).

Phylogenetic analysis and genomic characterization of PiBV

Phylogenetic analysis based on amino acid sequences from both segments of PiBV positioned the virus at the root of the genus Entomobirnavirus with high bootstrap support (Fig. 4). However, the identity of the full-length amino acid sequences of segment A and segment B to known Birnaviruses was below 30.9% and 35.5%, respectively, and the similarity was below 47.1% and 54.9%, respectively (Table 6).

Figure 4 Rootless phylogenetic trees of the segment A and B proteins of Birnaviridae.

The trees were constructed by the maximum likelihood method using the amino acid sequences for segment A (pVP2-VP4-VP3) and segment B (RdRp). A bootstrap probability with 1,000 replicates is shown for each node. The bold letters on the right are genera.

Table 6 Results of identity and similarity of amino acid sequences between PiBV and other members of Birnaviridae.

		SegmentA	SegmentB	
Genera	Spicies	GenBank acc. no.	Identity (%)	Similarity (%)	GenBank acc. no.	Identity (%)	Similarity (%)	
Entomobirnavirus								
	Mosquito X virus	JX403941	30.9	47.1	JX403942	35.2	53.5	
	Eridge virus	KU754527	30.8	46.5	KU754528	35.0	52.6	
	Culex Y virus	JQ659254	30.5	47.1	JQ659255	35.2	54.2	
	Espirito Santo virus	NC16518.1	30.1	46.8	NC_016517.1	35.1	54.2	
	Culicine-associated Z virus	KF298271	30.0	46.4	KF298272	35.1	54.9	
	Port Bolivar virus	MT263973	30.0	46.9	MT263974.1	35.5	54.1	
	Drosophila x virus	NC_004177	29.8	45.4	NC_004169.1	35.2	54.2	
Perbirnavirus								
	Largemouth bass birnavirus	MW727622	27.2	41.9	MW727623	29.9	44.0	
	Lates calcarifer birnavirus	NC_055452.	26.9	41.3	NC_055451.1	29.6	43.7	
Blosnavirus								
	Blotched snakehead virus	NC_005982	26.7	41.5	NC_005983.1	28.1	43.7	
Mambirnavirus								
	Porcine birnavirus	UDL09446	26.2	40.7	UDL09447	30.3	46.1	
Avibirnavirus								
	Infectious bursal disease virus	KU578104.1	25.5	37.3	KU578105.1	30.9	43.8	
Ronabirnavirus								
	Aedes birnavirus	MT381947	25.3	39.3	MT381950	30.4	30.4	
	Rotifer birnavirus	FM995220	25.2	41.0	FM995221	26.3	41.8	
Drnavirus								
	Drosophila melanogaster birnavirus	GQ342962.1	25.2	39.7	GQ342963.1	25.7	41.9	
Aquabirnavirus								
	Infectious pancreatic necrosis virus	AY283780.1	25.0	38.2	IPNVP1JAS	31.2	46.6	
	Victorian trout aquabirnavirus	NC_030242	25.0	40.8	KP268679	29.4	43.8	
	Yellowtail ascites virus	AB281673.1	24.8	38.6	AB281674.1	31.2	45.8	
Telnavirus								
	Tellina virus 1	AJ920335	24.6	41.3	AJ920336	24.2	39.7	

Each of the two segmental genomes of the Birnaviruses comprises the non-coding regions at both ends and a single open reading frame (ORF) in between (Delmas et al., 2019). The segment A of PiBV contains a 130-bp 5′ and a 95-bp 3′ untranslated regions, and a single ORF, while the segment B contains a 91 bp 5′ and an 85-bp 3′ untranslated regions, and a single ORF (Fig. 5). The GC content of PiBV’s segment A is 55.45%, and it does not possess the slippery UUUUUUAA motif that is conserved in Entomoviruses (Marklewitz et al., 2012; Huang et al., 2013; Tesh et al., 2020). The segment A encodes 1163-amino-acid pVP2-VP4-VP3, which is the second largest among Birnaviruses, after Aedes birnavirus (O’Brien et al., 2020). Birnavirus VP2, VP3 and VP4 conserved domains were found. Notably, the amino acid sequence of residues 571–590 of the pVP2-VP4-VP3 (GPPLKFWHFDFPDFVDVSDANN) shows 95% identity with the C-terminal region of the protein EP28 (GenBank: KT428641) from the pearl oyster. Segment B has a GC content of 54.02% and encodes a 988-amino-acid RNA-dependent RNA polymerase (RdRp, VP1).

Figure 5 Pinctada birnavirus genome organization.

Lines represent untranslated regions, boxes represent open reading frames (ORFs), and numbers below the genomes indicate both ends of the genome, the first nucleotide involved in the initiation codon, and the nucleotide at the ORF end. The ellipses depict predicted Birnavirus VP1-4 protein regions predicted by NCBI conserved domain search. Numbers in parentheses denote the respective amino acids. The sequence within the bold box exhibits homology to the C-terminus of the EP28 protein of the pearl oyster.

Epidemiological investigation

PiBV was detected in 14.3% to 100% of the pearl oysters of the groups that experienced summer atrophy but not detected in the pearl oysters from the areas with no reported outbreaks of the disease (Table 2).

Dynamics of PiBV in experimental infection for comparative metatranscriptomics

Figure 6 summarizes the results of RT-qPCR for the PiBV gene in the specimens sampled in the experimental infection. Despite that the nominal amount of 7.2 × 105 copies of PiBV gene was added to in each well in which each individual oyster was placed, much higher amounts of PiBV gene were detected in the specimens sampled on the second day of infection, suggesting the proliferation of the virus. The PiBV gene decreased rapidly thereafter, and it was not detected in three out of four oysters on the 21st day (Fig. 6). As shown in Fig. 3, the atrophy of soft body and death occurred later than PiBV proliferation.

Figure 6 The viral load in pearl oysters in the experimental infection for comparative metatranscriptomics.

Pearl oysters (of 4-month-old with the average shell width of 4.8 mm) were exposed to either the infection source or negative control inoculum for 2 h. Four oysters were randomly sampled from either the infection group (n = 48) or the control group (n = 48) on each sampling date. Only the data of the infected group are shown except for the initial four samples on day 0, which are not included in both groups. The numbers of the viral gene of all control oysters tested were under the limit of detection. Closed and open circles show copy numbers of the viral gene per individual. Open circles indicate copy numbers were below the detection limit. Crosses indicate means ± standard error. The incidence of atrophy and mortality in this experiment is shown in Fig. 2.

Immunohistochemistry

Positive IHC reactions for the PiBV protein were observed in two of three specimens sampled on days 2 and 4 from the infected group. No positive reactions were found in the other two oysters of the infected group; the mantle of these two animals were swollen and well extended, suggesting that these oysters were not severely infected with the virus. The mantle was apparently shrunken in two of the three oysters sampled on day 2 from the infected group. In these two specimens, positive reactions with anti-PiBV serum were observed mainly in the epithelial cells of the outer surface of the mantle (Figs. 7A–7D and 7F). The extrapallial space often contained positive cells that appeared to have been detached from the epithelium of the mantle (Figs. 7B–7D). Positive reactions were also observed in hemocyte-like cells in the mantle, adductor muscles (Fig. 7), or body cavity of other tissues. The mantle of the observed three individuals sampled on day 4 was swollen and extended toward the edge of the shell, although two of them showed positive reactions of the virus protein (Fig. 7G). Only small numbers of positive cells were observed on the outer edge of the mantle in the two oysters. Positive reactions were seen mainly in the detached cells in the extrapallial space and hemocyte-like cells inside the mantle (Fig. 7G).

Figure 7 Photomicrographs of the histological sections of experimentally infected juvenile pearl oysters with PiBV.

All panels except (E) are the results of immunohistochemistry (IHC) with mouse antiserum against the recombinant PiBV protein. Positive reactions appear reddish brown. (A–F) show different sections from the same individual sampled on day 2. (G) shows a section of a specimen sampled on day 4. g, gill; s, shell; m, mantle; am, adductor muscle; es, extrapallial space (the space between the shell and mantle); of, outer fold of the mantle; mf, middle fold of the mantle. (A) A section cut through the gills and adductor muscle. The hinge is on the right-hand side (not seen in this panel). Positive reactions (arrowheads) are seen mostly on the mantle. Some positive reactions are also observed in the swelled mantle tip (asterisk). Arrows indicate pigment of the pearl oyster. (B) An enlarged view of the mantle (arrows) surrounding the gills. Note that the mantle is very thin. Many cells with positive reactions are seen in the mantle epithelium, mantle cavity, and extrapallial space. (C) An enlarged view of the base of the adductor muscle. The arrowheads indicate the mantle. No positive cells are observed at the junction of the adductor muscle and the shell. (D) An enlarged view close to the outer edge of the mantle. Positive reactions are found in some mantle epithelial cells facing the extrapallial space. (E) The same area as shown in (D). This section is stained with hematoxylin and eosin. The sections shown in (D and E) are serial sections, but not adjacent to each other. It is difficult to identify PiBV-infected cells in this H&E-stained section. (F) An enlarged view of the outer edge of the mantle. The cells with positive reaction are mostly observed in the mantle epithelium facing the extrapallial space, which, in this section, is surrounded by the periostracum (arrowheads), outer epithelium of the mantle, and the shell. The periostracum is a thin proteinaceous sheet extending from the trough between outer and middle folds of the mantle (arrow) and covers the outer surface of the valve. (G) The panel shows one side of the mantle and shell of a day 4-sample. Note that the mantle is swollen as indicated by the arrows. Most of the cells with positive reactions are seen in the extrapallial space and inside the mantle.

Identifying virus-infected tissues was difficult in HE-stained sections. Even when the sites of immuno-positive reaction were observed in serially cut sections with H&E staining (Fig. 7E), cellular degeneration suggestive of viral infection, such as apoptosis or inclusion body formation, was not clearly observed. The only apparent histopathological change that suggests PiBV infection was the presence of many detached cells in the extrapallial space. However, it is not certain if the presence of these detached cells is pathognomonic of this disease.

Discussion

The results of the present study clearly suggest that PiBV is the causative agent for summer atrophy of the pearl oyster. Despite genes of many different viruses detected from both of the infection and control groups by metatranscriptomics, PiBV gene was detected only from the oysters of the infection group. The disease was successfully reproduced by the injection of purified fraction of PiBV. Although we could not see any visible bands of the virus nor observe virus-like particles with electron microscopy, a substantial amount of viral gene was detected in the purified fraction, but not in the negative control inoculum with RT-qPCR, while most of the other minor virus genes found in the purified fraction were also present in the control inoculum. Therefore, death and the melanin deposition observed in the purified fraction-injected groups can be attributed to PiBV. In addition, the viral gene was detected only from the areas where summer atrophy had been observed in the surveillance of pearl oyster culture farms. The diversity of viruses detected in the pearl oysters in metatranscriptomics is not surprising, as various viruses have been detected in bivalves (Meyers et al., 2009; Rosani & Gerdol, 2017; Matsuyama et al., 2017; Zhu et al., 2022; Jiang et al., 2023). Bivalves accumulate particles, including viruses, suspended in water because of their filter-feeding behavior (Zhu et al., 2022; Jiang et al., 2023; Carter, 2005; Errani et al., 2021; Volpe et al., 2023). Therefore, bivalves may contain not only infected viruses but also passively accumulated viruses (Volpe et al., 2023).

Although PiBV belongs to the family Birnaviridae, the low levels of identity and similarity suggest that the virus represents a novel genus. To date, several Birnaviruses have been reported to infect bivalves, including Tellina virus of the genus Telnavirus (Hill, 1976; Underwood et al., 1977; Nobiron et al., 2008), and various species of the genus Aquabirnavirus (Lo et al., 1988; Suzuki, Kamakura & Kusuda, 1998; Suzuki & Nojima, 1999; Kitamura et al., 2007; Meyers et al., 2009). However, PiBV is distinct from these groups. Interestingly, part of the viral genome showed high identity with the host protein, EP28. EP28 is a secretory protein involved in shell formation, which exists in the extrapallial fluid filling the space between the shell and mantle (Ji et al., 2010). To our knowledge, this is the first report of a virus in family Birnaviridae that shows sequence similarity with its host’s protein. Further studies are necessary to understand the virological significance of this similarity.

The mode of infection of PiBV revealed in this study is consistent with the symptoms of summer atrophy. The results of immunohistochemistry suggest that the primary target of PiBV is the outer epithelium of the mantle and other tissues are not severely affected. The hemocyte-like cells that exhibited positive reaction are likely to be the phagocytes that ingested debris of infected cells. As mentioned above, PiBV has a motif homologous to the extrapallial fluid protein involved in shell formation of the akoya oyster. Therefore, it is likely that the virus has taken up the gene fragment from the epithelial cells of the mantle, which secrete substrate proteins along with calcium carbonate to form shells (Sudo et al., 1997; Sato et al., 2013). The damage to the outer epithelium of the mantle caused by the virus infection most likely results in the contraction of the mantle and the atrophy of the soft body. Dark pigmentation or abnormal ridges of the nacreous layer of pearl oysters that have experienced summer atrophy is probably due to the altered position of the mantle, which was retracted toward the hinge, and also to the altered gene expression and disruption of the function of mantle epithelium affected by PiBV.

The present results, together with our previous report (Matsuyama et al., 2021), suggest that the infection of PiBV is acute and does not become chronic. The copy number of the virus genome peaked on the second day of the experimental infection and decreased thereafter. In addition, the animals sampled for histopathology from the infection group were already recovering from the disease 4 days after the start of the experiment. This may help to explain the low mortality of this disease in adult pearl oysters, although the disease often causes high mortality in juvenile oysters. Since the atrophy of the mantle probably results in the atrophy of the whole soft body including the gills, which is connected to the mantle, it seems plausible that some essential biological functions, such as respiration or food intake, are also impaired. Therefore, it is not surprising that PiBV infection can kill young oysters, which have weaker resistance to such impairment. Interestingly, the atrophy occurred when the viral load had already been declining. Deaths of the oysters were further delayed, starting on day 11 of the experimental infection (Fig. 3). This time lag of the appearance of clinical symptoms can be explained as follows: For contraction and atrophy, the mantle must be detached from the nacre surface of the shell. The detachment of the mantle from the shell most likely occurs when the infected mantle epithelial cells are dead and shed into the extrapallial space, by which time the active proliferation of PiBV has already ended. However, it would take time for the atrophied animals to recover, and young oysters that cannot endure the atrophied condition may die as discussed above.

Although mass mortality from this disease should be prevented at all costs, the acute and transient nature of the infection may make the diagnosis of the disease difficult. Compared to finfish farming, pearl oyster farmers observe their oysters less frequently. Therefore, an outbreak of the disease may be noticed after finding many empty shells or the abnormality of the nacreous layer of surviving oysters. There were probably many such cases, which is one of the reasons that made the identification of etiology of the disease difficult. This emphasizes the importance of normal sanitation practices, including not bringing animals from outside. However, when the disease was first noticed in 2019, the disease had already spread to most of the major farming regions of the pearl oyster in Japan (Matsuyama et al., 2021). This probably reflects the active transport of pearl oysters throughout Japan. Furthermore, pearl oysters or related species are often imported from foreign countries to Japan for the improvement of pearl quality. Unless these commercial practices are changed, prevention of the emergence and spread of new diseases would be difficult.

Conclusions

We identified a novel birnavirus as the causative agent of summer atrophy in the pearl oyster and named it PiBV. Histopathological analysis suggests that PiBV infects the mantle epithelium, resulting in atrophy. The small size (<100 nm) and the lack of envelope of the pathogen expected in previous infection experiments (Matsuyama et al., 2021) are consistent with those of Birnaviruses. Although PiBV is placed at the bottom of the genus Entomobirnavirus, a new genus should be proposed for PiBV because of its low similarity of amino acid sequence to other Birnaviruses. The RT-qPCR for a gene of PiBV and immunostaining developed in this study can be used for diagnosis and investigation to reduce economic losses in pearl farming due to summer atrophy.

Supplemental Information

Supplemental Information 1 Numbers of atrophied oysters and dead oysters observed daily during the experiment, and the cumulative mortality number and cumulative mortality rates.

Supplemental Information 2 The viral load in pearl oysters in the experimental infection for comparative metatranscriptomics.

*: Total RNA extracted from one individual was dissolved in 100 μL of DW, and the PiBV genome copy number in 1 μL of RNA solution was quantified.

†: The PiBV genome copy number per individual was obtained by multiplying the copy number in 1 μL of RNA solution by 100.

We thank Dr. S. Shirakashi, Kindai University, Mr. H Deguchi a member of the fishery cooperative association of Miura, Dr. M. Nakane, Kumamoto Prefecture Fisheries Research Center, Ms. M Murata, Nagasaki Prefectural Institute of Fisheries for their assistance for sampling pearl oysters.

Abbreviations

AbALV abalone asfa-like virus

IHC immunohistochemistry

MCP major capsid protein

NGS next generation sequencing

ORF open reading frame

PiBV Pinctada birnavirus

RdRp RNA-dependent RNA polymerase

RT-qPCR reverse transcription quantitative PCR

Additional Information and Declarations

Competing Interests

Author Contributions

Data Availability

The authors declare that they have no competing interests.

Tomomasa Matsuyama conceived and designed the experiments, performed the experiments, analyzed the data, prepared figures and/or tables, and approved the final draft.

Satoshi Miwa performed the experiments, analyzed the data, prepared figures and/or tables, authored or reviewed drafts of the article, and approved the final draft.

Tohru Mekata performed the experiments, analyzed the data, prepared figures and/or tables, authored or reviewed drafts of the article, and approved the final draft.

Ikunari Kiryu performed the experiments, authored or reviewed drafts of the article, and approved the final draft.

Isao Kuriyama performed the experiments, authored or reviewed drafts of the article, and approved the final draft.

Takashi Atsumi performed the experiments, authored or reviewed drafts of the article, and approved the final draft.

Tomokazu Itano performed the experiments, authored or reviewed drafts of the article, and approved the final draft.

Hidemasa Kawakami performed the experiments, authored or reviewed drafts of the article, and approved the final draft.

The following information was supplied regarding data availability:

The raw sequence data obtained in this research is available at the DDBJ Sequence Read Archive (DRA) under BioProject PRJDB15630.

The complete genome sequence of PiBV is available at GenBank: LC765995 and LC765996.

The contig sequences i. Table 3 that were generated by comparative metatranscriptomics are available at GenBank: ICTG01000001–ICTG01000027.

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
