# Peer review of "A novel birnavirus identified as the causative agent of summer atrophy of pearl oyster (Pinctada fucata (Gould))"

_PeerJ, doi:10.7717/peerj.17321_

## Round 0.1 · original submission · Major Revisions

This is an exciting manuscript - congrats on identifying the causative agent of this condition - always a challenge! All three reviewers were very positive in their assessment of this manuscript but provided a number of comments that need to be addressed to improve the revised manuscript. Note that you do not need a separate Discussion section, it is fine to have them combined.

·

Basic reporting

The research article is well written and describes a large body of well-planned research which was undertaken by the authors to identify the causative agent of summer atrophy which the authors previously described in PeerJ 2021;9: e12180 (reference 2 of this manuscript). It is refreshing to see a causative agent ascribed to this disease so soon after the disease was first identified. This has not been the case with several other suspected disease agents in Pinctada species such Akoya oyster disease, oyster oedema disease and “syndrome 85” where previously undescribed pathogens were suspected of having been introduced to a naïve host via translocation of molluscs from another location. Those diseases were referred to in the discussion of reference 2. Table 3 of the current manuscript which lists the contigs identified in the oysters with the best blast hits against known viral sequences may partially explain the difficulties in identifying potential pathogens in filter feeding molluscs where viruses are sequestered during feeding. The above aspects of the manuscript give it broader audience appeal than being solely about the identity of a novel pathogen in oysters affected by a recently described disease. There are quarantine and epidemiological implications of the research outlined in this manuscript.
The raw data, DNA data checks forming the custom checks, the tables and figures in the paper are easy to locate, well described, of good quality and comprehensive.

Experimental design

The Methods section of the paper describes in detail and in an orderly fashion the experimental design and techniques used for the research. The descriptions are comprehensive and provide validity to the conclusions. I would have liked a figure outlining the steps taken during the research to have been included because I found myself constructing one in order to better understand the sequence of events used during the research. For example the steps initially taken to obtain the “infection source”, also called the positive control in the line 196 of the “Pathogenicity of virus in purified fraction” section of the Methods, and later the “purified fraction” identified in line 186 of the “virus purification” section are important and could be easily outlined in a line diagram. Such a diagram could also clarify the fate of the 196 oysters put into the 1mL plastic wells in the Methods section on “experimental infection for comparative metatranscriptomics”.

Validity of the findings

The methods and results provide good evidence that a novel birnavirus is the causative agent of summer atrophy. The research provides evidence that enables Koch’s postulates to be fulfilled.

Additional comments

In the current manuscript the gross appearance of the oysters affected by summer atrophy are rather vaguely described as “soft body atrophy”, however reference 2 and Figures 1 and 5 of the current manuscript provide more comprehensive descriptions and images of the lesions.
The last paragraph of the Introduction is more like part of a Discussion. It could be re-worded to list the methods proposed to be used in the research to identify the causative agent.
The Results Section and raw data are comprehensive and describe the fate of all of the experimental oysters. It also provides information such as the number of virus particles identified in the “infection source” (positive control) and in the inocula injected into the oysters for the infection test Table 4) .
Some corrections that could improve readability are as follows:
• Line 249. “Each one specimen…” could read “One specimen from the infected and control groups….”
• Line 314. The sentence could say “DNA analysis was not performed because not enough DNA was recovered …..”
• Line 366. Delete the word “each”.
• Line 398. Replace “rather limited to“ by “mostly in “.
• Line 399. Replace “do” by “does”.
• Line 400. Replace “can explain” by “may help to explain”.
• Line 410. Replace “to bring” with “bringing”.
• Line 414. Replace “the quality of pearls” with “pearl quality”.
The Results and Discussion sectioned are combined in the manuscript but I do not consider this to be a major problem. There are some discussion-style comments in some section of the Results which is perhaps simpler than stating them in a general Discussion Section. However if a separate Discussion Section is required it could start at line 391.

·

Basic reporting

1. It is suggested to find a fluent English speaker to polish the language.
2. line338, The expression of this sentence is ambiguous: “Analyses of purified fraction and negative control inoculum with next generation sequencing”
2. The content of Table2 is inconsistent with the reference of Line315. It should be Table3. Table2 is out of order, the lines are not aligned, and number is miswritten.
3. Line 424-451 should be put in the discussion section.
4. Figure 2 and Figure 4 need to inform the number and the size of infected oysters and the method of infection in the figure notes. Also, the author also needs to make it clear in the paragraph of Line301~306

Experimental design

1. Please properly explain the difference between the results of line308: “Comparative metatranscriptomics” and line 338: “Analyses of purified fraction and negative control inoculum with next generation sequencing” in the Result part of the manuscript.

Validity of the findings

1. As the author said in Line 414-415: PiBV has a motif homologous to the extrapallial fluid protein involved in shell formation of the akoya oyster. Could the positive signal of immune hybridization (Figure 5) actually be the host protein? How to prove the specificity of hybridization results?
2. According to figure 2B, the death occurred on the 11th day. This is inconsistent with the results in figure 4. Oysters did not die in the early stages of active virus replication, but at the peak of death, virus replication war already very low. The author needs to explain this.

Additional comments

1. Line 365-379. it is better to have a schematic diagram of the structure of the virus genome. At the same time, it should also tell the readers how many fragments birnaviruses always have, and compare the structure of their genome.
2. Table 4, the author needs to clarify how positive control is defined and the size of the oysters used in the experiment.
3. Line 38: temperatures are high. So what is the specific temperature?

·

Basic reporting

Matsuyama and colleagues proposed the paper titled “ A novel birnavirus identified as the causative agent of summer atrophy of pearl oyster (Pinctada fucata (Gould))” focused on the identification of the causative agent of pearl oyster summer atrophy. The authors used meta-transcriptomic data to identify a novel virus, called PiBV, as the possible causative agent. They substantiate this hypothesis with relevant data, supporting this finding appropriately.

Experimental design

Appropriate

Validity of the findings

Valid, although some parts of the results could be ameliorated.

Additional comments

Although the overall methodology is sounding, I have several points that in my opinion should be addressed before considering the paper for publication in PeerJ.
Methodological details regarding data analysis are not well described or appropriate. For examples, the authors applied a filter based on the number of mapped reads to select relevant contigs. However, a normalization based on contig lengths is required to take into consideration the very different length range of the obtained ocntigs (e.g. TPMs or RPKMs). Although the authors stated that they use CLC for the analyses, some more parameters are needed (e.g. mapping parameters). Perhaps, the use of a more accurate assembler is preferred since the CLC assembler, although very fast, does not provide results accurate as others, possibly having designed pipelines for RNA viruses (e.g. SPADES). Also the statements that for viral identification the viral sequences deposited in database have been used is too vague. Here the use of a more sensitive method for viral identification could provide more information (e.g. using HHMer with the RdRP as bait or palmscan). Instead of the number of mapped reads, a value considering the genome length should be used also in the tables (e.g. TPMs).
This statement “From the results of comparative metatranscriptomics, we identified a birnavirus, designated as PiBV,” should be better circumstantiated or removed from the M&M section.
In the part regarding phylogenetic analysis, the methods are not enough described (e.g. what parameters were used to filter out the alignment with Gblocks?).
The paragraph titled “Dynamics of PiBV in experimental infection for comparative metatranscriptomics” could be joined somewhere else since it is very short and poorly informative.
In the results, apart from an update based on the methodological suggestions, I would consider as informative a visualization of the viral genome with coding regions, UTRs and so on.
The section “Discussion” is not present as standalone section. The peculiar phylogenetic position of this virus may arose from the lack in database of similar viruses infecting bivalves? Have to authors tried to retrieve these sequences?

---

## Round 0.2 · accepted · Accept

Thank you for addressing the reviewers' concerns, this manuscript is now ready for publication.

·

Basic reporting

no comments

Experimental design

no comments

Validity of the findings

no comments

Additional comments

no comments